# The cryo-EM structure of hibernating 100S ribosome dimer from pathogenic *Staphylococcus aureus*

Donna Matzov[1], Shintaro Aibara[2], Arnab Basu[3], Ella Zimmerman[1], Anat Bashan[1], Mee-Ngan F. Yap [3], Alexey Amunts [2] & Ada E. Yonath[1]

Formation of 100S ribosome dimer is generally associated with translation suppression in bacteria. *Trans*-acting factors ribosome modulation factor (RMF) and hibernating promoting factor (HPF) were shown to directly mediate this process in *E. coli*. Gram-positive *S. aureus* lacks an RMF homolog and the structural basis for its 100S formation was not known. Here we report the cryo-electron microscopy structure of the native 100S ribosome from *S. aureus*, revealing the molecular mechanism of its formation. The structure is distinct from previously reported analogs and relies on the HPF C-terminal extension forming the binding platform for the interactions between both of the small ribosomal subunits. The 100S dimer is formed through interactions between rRNA h26, h40, and protein uS2, involving conformational changes of the head as well as surface regions that could potentially prevent RNA polymerase from docking to the ribosome.

[1] Faculty of Chemistry, Department of Structural Biology, The Weizmann Institute of Science, Rehovot 7610001, Israel. [2] Science for Life Laboratory, Department of Biochemistry and Biophysics, Stockholm University, 17165 Solna, Sweden. [3] Edward A. Doisy Department of Biochemistry and Molecular Biology, Saint Louis University School of Medicine, St. Louis, MO 63104, USA. Donna Matzov and Shintaro Aibara contributed equally to this work. Correspondence and requests for materials should be addressed to F.Y.M.-N. (email: myap1@slu.edu) or to A.A. (email: amunts@scilifelab.se) or to E.Y.A. (email: ada.yonath@weizmann.ac.il)

The ribosome is the universal cellular ribonucleoprotein assembly that translates the genetic code into proteins. Suppressing the translation process under stress conditions is an adaptation mechanism, as protein synthesis is a highly energy-consuming function[1–4]. Gammaproteobacteria such as *Escherichia coli* possess three ribosome-silencing factors that bind to the small ribosomal subunit and block the binding pocket of messenger RNA (mRNA) and the anticodon region of A-, P-, and E-transfer RNA (tRNA)[5,6]. YfiA promotes the inactivation of the 70S, as it prevents ribosomal recycling for translation initiation[7,8], ribosome modulation factor (RMF), and hibernating promoting factor (HPF), which concertedly induce the dimerization of two 70S ribosomes into a translationally inactive, hibernating 100S complex[9–11]. *E. coli* 100S dimers are mostly found during the stationary growth phase when nutritional sources are scarce[12–14]. This process is reversible, as once fresh nutrient sources are introduced, the 100S dimers can dissociate within minutes back into active 70S ribosomes[13,14]. In vitro studies and cryo-electron microscopy (cryo-EM) showed that the two 70S ribosomes are conjoined together through their small subunits in a 'head-to-head' configuration to form the 100S complex[15–17].

Ribosomes of Gram-positive bacteria such as *Staphylococcus aureus* also form 100S dimers. However, they can be found throughout all growth phases, even when nutrients are ample[15,18–21]. The significance of the existence of hibernating 100S during the exponential growth phase has not been completely understood. Recent studies suggested that the formation of 100S ribosomes in the logarithmic growth phase of *S. aureus* reduces the translation efficiency of a fraction of genes[22]. Similarly, it was demonstrated that loss of *S. aureus* HPF (HPF$_{SA}$) causes massive ribosome breakdown upon entering the stationary phase that correlates with the onset of cell death[22]. In addition, deletion of HPF in *Lactococcus lactis* exhibited decreased viability after resuscitation from starvation conditions[17] and HPF-depleted *Bacillus subtilis* cells ability to regrow from the stationary phase was decreased[21]. Finally, deletion of *hpf* in *Listeria monocytogenes* impaired the survival of the bacteria in a murine model of infection[23] and compromised its tolerance to aminoglycosides[24].

In contrast to *E. coli*, *S. aureus* does not carry RMF or YfiA homologs. Instead, it contains only an *hpf* gene that encodes another form of HPF$_{SA}$, which is twice as long as HPF$_{EC}$[9,25]. This long form of HPF is common among all Gram-positive bacteria and plant plastids (Supplementary Fig. 1)[17,20,23,26]. To reveal the basis for ribosome dimerization in *S. aureus*, we have determined the structure of the native 100S complex (SA100S) using single particle cryo-EM. The structure revealed that the two 70S ribosomes intertwine intricately through their 30S subunits and these interactions involve ribosomal protein uS2, ribosomal RNA helices h26 and h40, and the C-terminal domain of HPF$_{SA}$ (C-HPF$_{SA}$). A loss of dimerization sensitizes cells to heat stress. In addition, the N-terminal domain of HPF$_{SA}$ (N-HPF$_{SA}$) blocks the mRNA and the anti-codon tRNA binding sites, demonstrating the dual functionality of the HPF$_{SA}$. Finally, the structure of the *S. aureus* 100S dimer was found to be distinct from its analogous *E. coli* complex and therefore may be a potential species-specific therapeutic target.

## Results

### Structure determination of the SA100S complex.

For cryo-EM analysis, 100S ribosome dimers were isolated and purified from a *hpf*-knockout *S. aureus* strain (USA300) harboring a high-copy plasmid encoding for the *hpf* allele. Tenuous density corresponding to one of the 70S monomers in the two-dimensional (2D) class averages suggested that the connection between the two 70S ribosomes is generally flexible. This was confirmed by

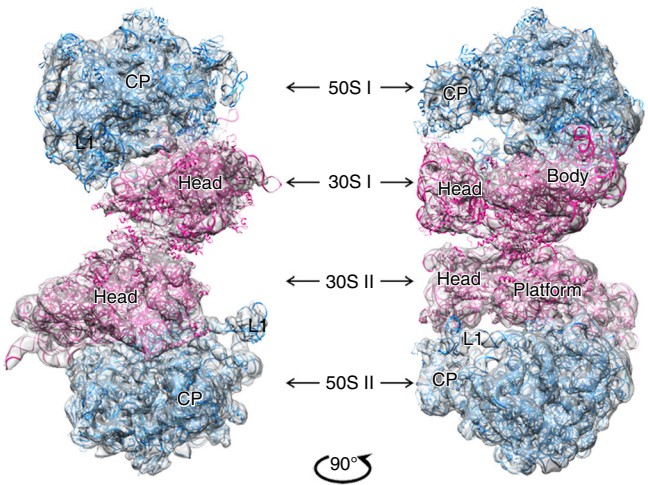

**Fig. 1** Overall structure of the SA100S ribosome dimer. The paired ribosomes within the 6.8 Å cryo-EM map of the 100S are connected to each other through the 30S subunits (colored in *pink*) and are oriented in a twofold symmetry relative to each other. The 50S subunits are colored in *blue*. Domain labels are: CP, central protuberance; L1, uL1 arm; Head, 30S subunit head; platform, 30S subunit platform; Body, 30S subunit body

computational sorting of the three-dimensional (3D) classes, where one ribosome was typically sharper than its interacting counterpart. To overcome the conformational flexibility of the 100S, we collected a large data set of 12,504 micrographs and used in silico classification to select a distinct homogeneous sub-population of a sufficient number of particles with RELION 2.0[27] (Supplementary Fig. 2). Consequently, a single 3D class of 12,570 particles, representing 5.86% of the total particles, with a well-defined orientational state for both 70S ribosomes, was obtained. These particles were further refined and reconstructed to a 8.2 Å resolution cryo-EM map. Inspection of the two 70S copies in the asymmetric reconstruction suggested that there was a clear axis of twofold symmetry within the 100S particle in agreement with the previous studies[15,16]. Thus, a refinement with C2 symmetry was applied and improved the overall resolution to 6.8 Å, sufficient for molecular interpretation (Supplementary Figs. 2 and 3). Features of the ribosomes were readily identified in this map and coordinates of the 70S were fitted (Fig. 1).

In order to examine the detailed structure of HPF$_{SA}$ and its interactions with the ribosome, the data were re-processed as monomers, i.e., the 70S ribosomes composing the dimeric particles were treated individually, which yielded a 3.0 Å cryo-EM density map for the 70S particle (Supplementary Fig. 4). To further improve the quality of the maps and to permit accurate model building, the small and large subunits were refined separately by applying a mask on the specific subunit during its refinement. This resulted in a 3.2 and 2.9 Å resolution maps of the 30S and 50S subunits, respectively, which allowed accurate building of an atomic model of both subunits into their corresponding maps.

### C-HPF domain provides binding platform for 30S dimerization.

The first 100 residues that form the N-HPF$_{SA}$ were clearly identified and modeled using the masked 30S density map (Supplementary Fig. 5). This structure revealed that the N-HPF$_{SA}$ is involved in several interactions with the 16S rRNA (Fig. 2a) and it is bound between the head and the body of the small ribosomal subunit in a pocket that is composed of rRNA helices h18, h23, h25, h28-h31, h34, h44, and ribosomal proteins uS7, uS9, and uS11. Superposition of *Thermus thermophilus* 70S (*Tth*70S)

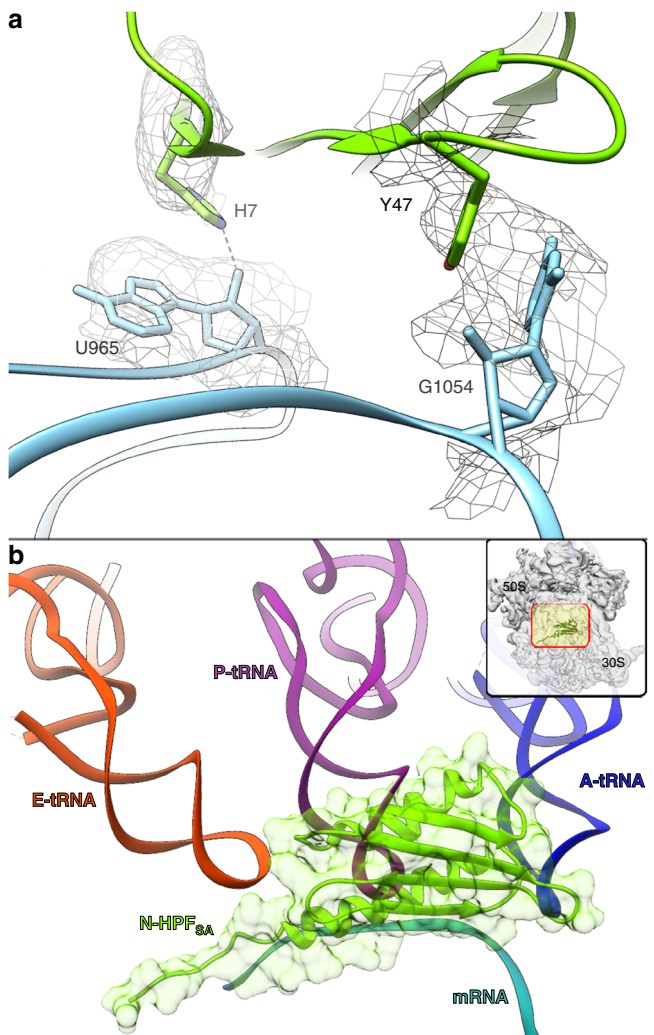

**Fig. 2** Structure of the N-terminal domain of HPF. **a** The N-HPF$_{SA}$ (*chartreuse*) is interacting with the rRNA (*light blue*) in several locations. For example, H7 form hydrogen bonds with U965 (*E. coli* numbering) and Y47 is stacked with the base of G1054. **b** Superposition of *Tth*70S complex with mRNA (*cyan*) and three tRNAs (A-site in *blue*, P-site in *magenta* and E-site in *orange*, PDB code 4W2F) on one of the 70S ribosome composing the SA100S dimer presented here, demonstrating that the N-HPF$_{SA}$ blocks the binding sites of mRNA and the anti-codon region of all three tRNAs. The N-HPF$_{SA}$ location on a single 70S monomer is shown in the top right inset

complex with mRNA and three tRNAs at their binding sites (PDB code 4W2F[28]) on one of the 70S ribosomes composing the SA100S dimer presented here demonstrates that N-HPF$_{SA}$ physically blocking the binding pocket of mRNA and tRNA (Fig. 2b). The first step of prokaryotic protein synthesis involves the binding of a specific initiator methionyl tRNA and mRNA to the small ribosomal subunit to form an initiation complex. The large ribosomal subunit then joins the complex to form an elongation-competent ribosome. The adaptor function of the tRNAs involves two distinct regions of the molecule. All tRNAs possess CCA at their 3′-terminus and amino acids are covalently attached to the ribose of the terminal adenosine, whereas the mRNA template is recognized by the anticodon loop, located at the other end of the tRNA through complementary base pairing. Hence, blocking the binding pockets of the mRNA and tRNA molecules inhibits the translation process. The sequences of the N-terminal 100 residues of HPF$_{SA}$ and the short HPF$_{EC}$ have 29% identity, with high structural conservation (Supplementary Fig. 1). RMF$_{EC}$ also shares

a similar degree of identity with the middle part of HPF$_{SA}$; therefore, it was presumed that *S. aureus* long HPF might be a hybrid of the short HPF and RMF. However, superposition of the crystal structure of *Tth*70S in complex with RMF$_{EC}$[5] to this HPF$_{SA}$-70S map, revealed no extra density at RMF$_{EC}$-binding pocket or at its vicinity (Supplementary Fig. 6).

To further improve the density at the 100S ribosome dimer interface, we performed a focused refinement on this region with both 50S subunits masked out and applying C2 symmetry. This yielded a 6.7 Å resolution map of the dimer interface (Supplementary Fig. 2). The improved density allowed interpretation of the interaction network between the two 30S subunits. In our 100S model, protein uS2 and helix h26 of the 16S rRNA are intertwined so that h26 of one ribosome is interacting with uS2 of the other ribosome and vice versa (Fig. 3a, b). At the interface of the two subunits an additional density was observed (Supplementary Fig. 7). This density was also characterized by a twofold symmetry, similar to the twofold between the two 70S ribosomes that form the 100S dimer, hence improved when C2 symmetry was applied (Supplementary Fig. 2). Given that all the ribosomal proteins were built, as well as the N-HPF$_{SA}$, we assigned this density to the C-HPF$_{SA}$. Homology modeling based on the available crystal structure of a ribosome-associated factor Y from the Gram-positive bacteria *Streptococcus pyogenes* (PDB ID: 3LYV) and on biochemical assays (Supplementary Fig. 7D–H), we suggest dimerization of the C-HPF$_{SA}$ (Supplementary Fig. 7C). The local resolution of ~ 6 Å in this region allowed fitting the C-HPF$_{SA}$ dimer at the 100S interface, showing that it further interacts with helix h40 of the adjacent 30S (Figs. 3a, b). Owing to the limited resolution of the cryo-EM maps in this region, the specific interactions between the two C-HPF$_{SA}$ domains that allows self-dimerization could not be characterized. However, the homology model suggests a 3D domain-swapping mechanism in which the β-strand of one C-HPF$_{SA}$ is packed in parallel against the antiparallel β-sheet of its dimer counterpart (Fig. 3c). The fact that the intra β-strand interaction is parallel to the β-sheet makes the dimer less stable.

To confirm the cryo-EM structure observation, we constructed *S. aureus* strains harboring the N-terminal domain (NTD, residues 1–100) and C-terminal domain (CTD, residues 121-190). Although ribosomes purified from the NTD strain lose the ability of forming the 100S dimer, the CTD strain ribosomes behave similarly to the strain harboring the full-length HPF$_{SA}$ (Supplementary Fig. 7D, E). In addition, in vitro assays demonstrated that the full-length HPF$_{SA}$ protein as well as the C-HPF$_{SA}$ construct are able to self-dimerize (Supplementary Fig. 7F–H). The 70S dimerization is critical for stationary phase cell survival under acute heat stress. At non-permissive temperature (62 °C), the empty vector control and NTD strains were > 100-fold more susceptible to heat killing than the full-length HPF and CTD strains (Supplementary Fig. 8). Taken together, our data suggest that C-HPF$_{SA}$ forms the binding platform for the adjacent 30S, which is essential for *S. aureus* 100S dimerization. Furthermore, the superposition of our structure and the native SA70S[19] revealed that the presence of HPF$_{SA}$ induces a conformational change of helices h26 and h40 (Fig. 3d). Previous studies demonstrated both in *S. aureus* and *L. lactis* that the 100S dimers cannot be formed without the presence of HPF[17, 22]. Therefore, we suggest that upon HPF$_{SA}$ binding a conformational change in h26 and h40 occur, which allow them to interact with uS2 and C-HPF$_{SA}$ to further stabilize the dimer.

**30S conformational changes is induced by 100S stabilization.** Ribosomal 30S head is a highly dynamic domain able to adopt major swiveling rearrangements. To reveal the degree of relation between

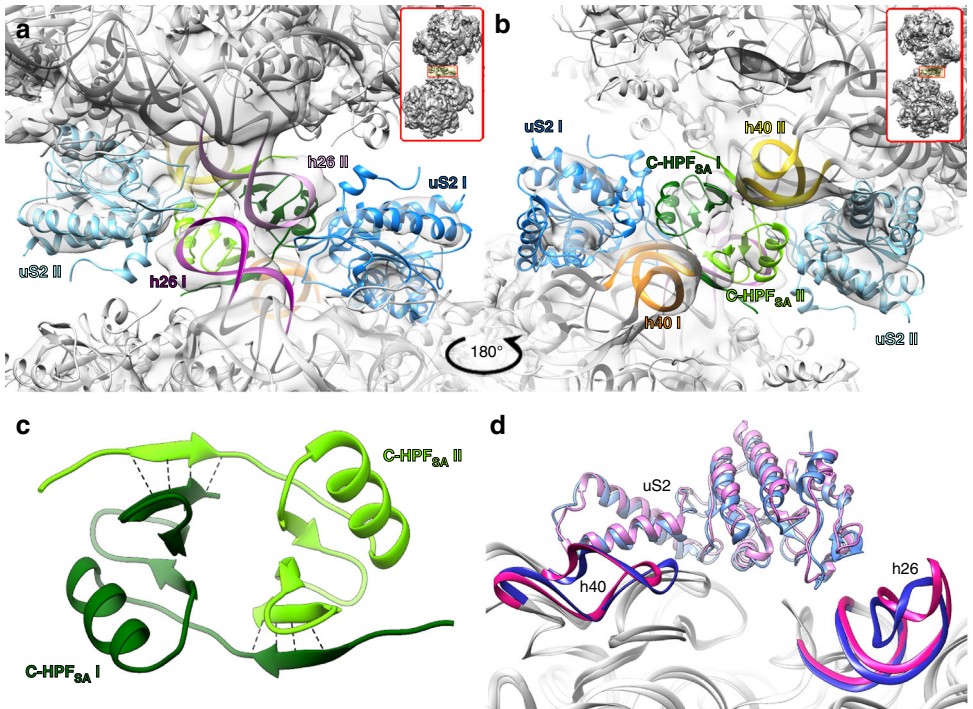

**Fig. 3** *Close-up view* of interface between the two 30S subunits. **a** h26 of the top ribosome (*pink*) is interacting with uS2 of the bottom ribosome (*blue*) and vice versa (h26 in *magenta* and uS2 in *light blue*). **b** h40 of the top ribosome (*yellow*) is interacting with C terminal HPF$_{SA}$ (C-HPF$_{SA}$) of the bottom ribosome (*dark green*) and vice versa (h40 in *orange* and the C-HPF$_{SA}$ in *light green*). View of the whole 100S and the interface region within the 100S structure of both **a** and **b** is shown in the upper inset. **c** Homology model of the C-HPF$_{SA}$ suggests the two domains self-dimerized via β-sheet interactions. **d** Superposition of SA100S and *S. aureus* 70S (SA70S PDB code 5LI0). Helices 26 and 40 are in *blue* and *hot pink*, respectively. Ribosomal protein uS2 of SA100S is in *light blue*, uS2 of SA70S is in *light pink*, and the rRNA of both structures is in *gray*

the 100S dimerization and the head conformational changes, we used focused masked classification over HPF$_{SA}$ and refined ribosomes as monomers (Supplementary Fig. 3). Comparison of the distinct 3D classes showed that upon HPF$_{SA}$ binding a concerted movement of the head and the body occurs, similar to the ratcheting motion of the head during translocation. The head of the 30S subunit rotates with respect to the body by an angle of about 5°, resulting in a maximum ~ 3 Å displacement of the shoulder helix h16 and ~ 8 Å displacement of the beak helix h33 (Fig. 4a and Supplementary Movie 1). The rotation is counterclockwise when seen from the solvent side of the 30S subunit and results in decreased contact surface between the head domain and the body.

According to the structures of *Tth*70S in complex with RMF$_{EC}$ and HPF$_{EC}$ (PDB code 4V8G and 4V8H, respectively), upon binding of RMF$_{EC}$ the head of the 30S subunit undergoes a conformational change, which facilitates the formation of the 100S dimer[5]. A similar conformational change was observed upon HPF$_{EC}$ binding to *Tth*70S. However, as shown above, the N-HPF$_{SA}$ that is structurally similar to HPF$_{EC}$, is not sufficient by itself to form 100S dimers in *S. aureus*. Thus, it seems that the conformational change induced by HPF$_{SA}$ binding stabilizes the conformation of uS2 and enables participation in dimer formation. Indeed, protein uS2 is better resolved in our maps where HPF$_{SA}$ is bound compared with apo-ribosomes.

**Distinct structural differences between SA100S and EC100S.** *S. aureus* 100S complex formation involves interactions between protein uS2 and helices h26 and h40 of the 16S rRNA. Two modes of *E. coli* 100S (EC100S) complex formation have been suggested according to two independent cryo-EM studies; In the first, the ribosomal proteins uS2, uS3, and uS5 of the 30S are involved in dimer formation[15], whereas in the second, uS2, uS9,

uS10, and helix h39 are involved in this process[16]. In the SA100S, only the head (uS2 and h40) and platform (h26) of the 30S subunits take part in the dimerization process, whereas in the EC100S, the body (uS2, uS9, uS10, and h39) participates too in 100S dimerization. Alternatively, Kato et al.[15] have proposed that the body (uS5) domain of the 30S is also part of the EC100S formation. Superposition of our SA100S with EC100S maps (EMDB 1750, 5174) through alignment of one of the paired 70S, revealed significantly different conformations of the dimer formed between the two 30S subunits (Fig. 4b, c). In either EC100S reconstruction, one 70S ribosome is rotated with respect to its counterpart in the SA100S by an angle of ~ 110° such that the interface between the two 30S in EC100S spans over a much larger surface area. Therefore, the essence of interactions forming the 100S and the structural elements involved is not universal, but species dependent. In *S. aureus*, rRNA helices h26, h40, and ribosomal protein uS2 form the dimerization interface, whereas in *E. coli*, rRNA helix h39 and proteins uS9, uS10 or uS2, uS3, and uS5 are involved in a more expanded interaction surface.

## Discussion

The 100S complex was first identified in bacteria over 50 years ago[29, 30]. Extensive biochemical studies were performed since then, in order to identify which factors take part in the dimer formation, as well as understanding the underlying molecular mechanism and significance of this phenomenon. Two distinct types of bacterial 100S ribosomes have been identified; the first in Gammaproteobacteria and Betaproteobacteria during the stationary growth phase that is induced by two proteins, short HPF, and RMF. The other type of 100S dimer is typical for Gram-positive bacteria and is present throughout all growth phases. Structural studies performed using *T. thermophilus* 100S, revealed

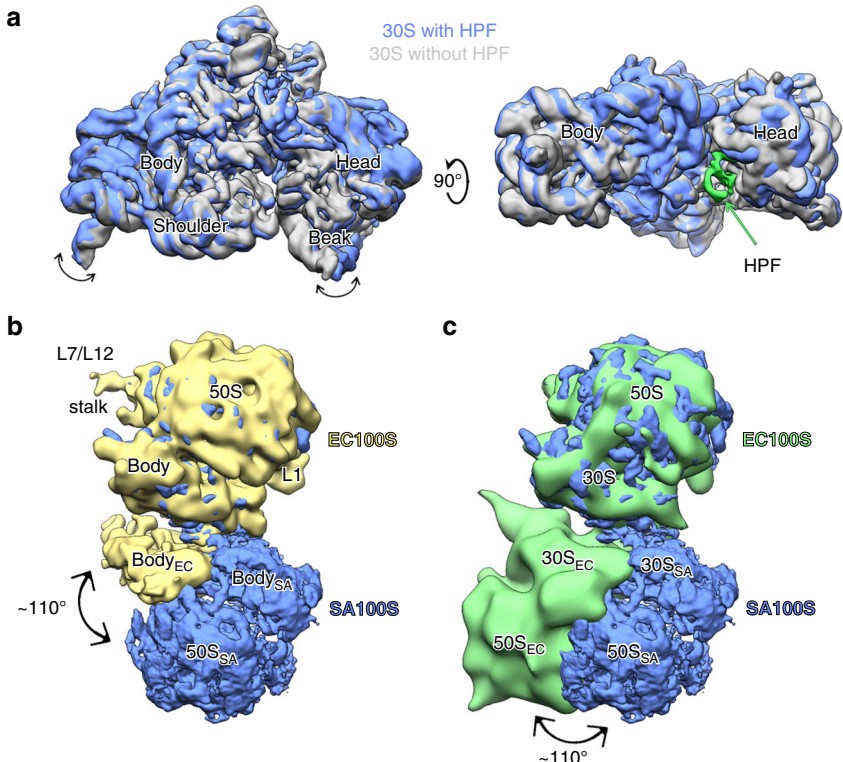

**Fig. 4** Structural differences of C-HPF$_{SA}$ bound/unbound 30S subunits and *S. aureus* vs. *E. coli* 100S structures. **a** Superposition of two small ribosomal subunits; bound and unbound to HPF from two views demonstrates conformation change upon C-HPF$_{SA}$ binding. The bound 30S is in *blue* and the unbound is in *gray*. The density of C-HPF$_{SA}$ is shown on the *top panel. Arrows* demonstrate the movement directions of the 30S head and body upon binding. **b**, **c** Superposition of SA100S (in *blue*) on **b**. Single-particle cryo-EM map of part of EC100S (in *yellow*, EMDB code EMD-1750) and **c** cryo-EM tomogram of EC100S (in *green*, EMDB code EMD-5174)

how *E. coli* HPF and RMF interactions with the heterologous ribosome enable its hibernating form. Cryo-EM density maps reconstructed from *E. coli* 100S particles revealed that the two ribosomes composing the dimer are connected through their small subunit. However, due to their low resolution, the nature of these interactions has remained ambiguous. Although parts of the 30S dimerization machinery in *E. coli* have been described; the formation of 100S dimers mediated by long HPFs found in Gram-positive bacteria is even less understood.

In this study, the initial 6.8 Å density map reconstructed from single particles of 100S ribosomes of *S. aureus* allowed a more accurate fitting of an atomic model, which revealed that similar to EC100S dimers, the ribosomal pair of the SA100S is bound through their small subunits. The 30S–30S interactions are mediated by a homo-dimer of CTDs from two HPF$_{SA}$, which allows for rRNA and ribosomal proteins to interlock and further secure the connection between the two 70S ribosomes. Such dimerization presents two recognition sites for rRNA binding, in this case h26 form both 30S subunits and can therefore provide a cooperative interaction that strengthens the affinity of the CTD for the rRNA[31]. Notably, although the interactions between the two 70S should be strong, they certainly are not rigid, as the 3D classification shows that the large majority of the dimers exhibit flexibility between to two ribosomes. Indeed, the dimer stabilizing parallel β-sheet interactions facilitate weaker dimer interaction (compared with an antiparallel arrangement).

Re-processing the cryo-EM data and reconstructing density maps from individual 70S ribosomes that composed the dimer yielded density maps at a resolution of 2.9–3.2 Å that allowed us to trace the N-HPF$_{SA}$. The binding site of N-HPF$_{SA}$ overlaps with that of the mRNA and the tRNA anti-codon loop, thus explaining how HPF halts protein synthesis by steric block of the decoding site. Although the NTD is well resolved within the high-resolution maps, no density was found to be connecting between this domain and the CTD. This may indicate that the two domains are linked via an unstructured, highly flexible loop. These observations alongside sucrose density analysis and negative stain of HPF-truncated strains led to the conclusion that HPF$_{SA}$ has dual functionality. The NTD is responsible for silencing the ribosomes, whereas the CTD is directly involved in the dimer formation. As the NTD by itself is sufficient for silencing the ribosome, much similar to the protein YfiA, the biological significance of the ribosome dimerization remains unclear. It is possible that the 30S–30S dimerization interface precludes the accessibility of RNases, because the abundance of SA100S is inversely correlated with the degree of ribosome degradation when nutrient is depleted[22]. Under heat stress, dimerization appears to have a protective role in mitigating ribosome damage that is manifested by cell death (Supplementary Fig. 8). Furthermore, bacterial translation and transcription are coupled and the binding sites (e.g., uS2) of RNA polymerase (RNAP) on the 70S ribosome[32] coincidently overlap with the 70S dimerizing interface. Therefore, one might speculate that homo-dimerizing 70S prevent spurious interactions between the RNAP and ribosome in the crowded cytoplasm when translation should not be occurring.

Even though the 100S complexes of gammaproteobacteria and Gram-positive bacteria serve the same purpose, the structural basis for dimerization is different. It is quite likely to be that all bacteria carrying a long HPF protein form 100S complexes in the same manner presented here. Hence, hampering the dimer formation by targeting their signature interface between the two ribosomes may offer a unique Gram-positive specific anti-bacterial treatment.

## Methods

**Bacterial strains and growth conditions**. *S. aureus* strain JE2 and its *hpf*-null mutant[22] are derivatives of the community-associated methicillin-resistant USA300 isolate (NR-46543, BEI Resources)[33]. Restoration of the 100S ribosome formation in the *hpf*-null background was tested with the complementing plasmids harboring either the full-length *hpf* (pLI50hpf), CTD deletion (pLI50hpf (1–100)), or NTD deletion (pLI50hpf (121-190)). The full-length *hpf* was PCR amplified with primers P651 (5′-CGG GAT CCA TAC AAC TGG ATT AAC AAT TCA TCG TGC AGG GTG-3′) and P627 (5′-TGA AGC TTT AAA CTT AAT TTA TTG TTC ACT AGT TTG AAT CAA GCC-3′) and the genomic DNA of JE2 as a template. A PCR fragment corresponding to the 1-100 amino acids of *hpf* was amplified with primers P651 and P722 (5′-TGA AGC TTT ACT TAT CGA TTA ATA CGT GTT TTA TAT TTT CG-3′). To obtain *hpf*(121–190) region, the NTD deletion was introduced by two-step crossover PCR[34] using four primers (P651; P1080 (5′-ATC ATC GTA AGC GTC ATT ATC CAT AGT AAT CTC TCC TTA AAC CTC TTT AT-3′); P1079 (5′-ATA AAG AGG TTT AAG GAG AGA TTA CTA TGG ATA ATG ACG CTT ACG ATG AT-3′);P627). The primer pairs P651/P1080 and P1079/P627 were used in the first PCR reaction with the JE2 genomic DNA as a template. The P651/P627 pair was used in the second reaction using PCR products of the first reaction as the templates. The *hpf* PCR fragments were then cloned into the *Bam*HI and *Hind*III sites of pLI50[35], in which the expression of *hpf* variants is under the control of its native promoter and Shine-Dalgarno sequence. Plasmids were passaged through the restriction-deficient *S. aureus* strain RN4220[36]. The plasmids were isolated from strain RN4220 and transformed into the *S. aureus* JE2Δ*hpf* mutant. *S. aureus* strains carrying pLI50 derivatives were routinely grown in tryptic soy broth (TSB, BD Difco) supplemented with chloramphenicol (10 μg ml$^{-1}$).

The overexpression and purification of recombinant His-tagged full-length HPF from pET28a (Novagen) have been previously described[22, 37]. To overexpress the HPF-CTD, heptahistidine tag was incorporated to the N terminus of CTD by PCR amplification using primers P1076 (5′-TAC CAT GGG CCA TCA TCA CCA TCA CCA TCA TAG CAG CGG CAT AGA AAT TAT TCG TTC AAA-3′) and P1078 (5′-CGC TCG AGT TAT TGT TCA CTA GTT TGA ATC-3′). The DNA fragment was cloned into the *Nco*I and *Xho*I sites of pET28a, yielding plasmid pEThpf (121–190). A mini Ser-Ser-Gly linker was introduced downstream of the heptahistidine preceding the 121 residue of *hpf*. *E. coli* BL21(DE3) bearing pET28a derivatives was grown in LB supplemented with 50 μg ml$^{-1}$ kanamycin and induced with a final 0.5 mM isopropyl β-D-1-thiogalactopyranoside at cell density OD$_{600}$ ∼ 0.45. All strains were cultured at 37 °C unless otherwise noted.

**Ribosome purification**. Cells were grown in TSB medium to an OD$_{600}$ of 0.7–0.8 and collected by centrifugation at 11,540 *g* for 30 min at 4 °C. The collected cells were immediately suspended in ice-cold buffer A (20 mM Hepes-KOH (pH 7.5), 14 mM Mg$^{2+}$ acetate, 100 mM KCl, 1 mM dithiothreitol and 0.5 mM phenylmethylsufonyl fluoride). The suspended cells were flash frozen as bacterial spheres by slowly pipetting the slurry into liquid nitrogen and stored at −80 °C. To disrupt the cells membrane, the frozen bacterial spheres were cryo-milled into a fine powder, which was then suspended in buffer A. The cells debris was discarded by centrifuging the lysate at 30,000 *g* for 20 min at 4 °C. The cell extract was collected and layered onto a 1.1 M sucrose cushion in buffer A and centrifuged at 310,000 *g* for 17 h at 4 °C. The ribosomal pellet was re-suspended in buffer A and layered onto a 10–40% sucrose gradient and centrifuged using a SW-28 rotor, at 18,000 r.p.m. for 17 h at 4 °C. The desired fractions were collected, combined, and centrifuged overnight at 320,000 *g*, 4 °C. The ribosomal pellet was re-suspended in buffer B (20 mM Hepes-KOH, 10 mM Mg$^{2+}$ acetate, 100 mM KCl) and centrifuged again, using a TLA-100.2 rotor at 75,000 r.p.m. for 1.5 h at 4 °C. The ribosomal pellet was re-suspended in buffer B, diluted to a concentration of ∼ 1mg ml$^{-1}$, aliquoted, and flash frozen in liquid nitrogen.

**Cryo-EM data collection and refinement**. For the data set used to obtain the high-resolution structure of the 70S ribosome, 3 μl of purified ribosomes (OD$_{260 nm}$ = 3.0) were applied on to freshly glow discharged holey carbon grids (Quantifoil R2/2 Cu) pre-coated with a home-made continuous carbon film (∼ 30 Å thick) and incubated for 30 s at 4 °C, 100 % humidity in a FEI Vitrobot Mk IV system. The grids were blotted for 3 s before plunge cooling in liquid ethane. In the case of the 100S ribosome dimer, three data sets were collected over several non-consecutive days and combined in attempts to minimize orientation bias. To pick dimers confidently, a more diluted sample (OD$_{260 nm}$ = ∼ 2.2) was used, and blotting and freezing parameters were kept the same as above.

Micrographs were acquired using an FEI Titan Krios electron microscope operated at 300 kV. Images were acquired using FEI's automated acquisition software (EPU) using a FEI Falcon II detector at a pixel size of (1.07 Å per pixel). All data sets were acquired at 17 frames per second and the total dose used was 25 electrons per Å$^2$. Movies were processed using Motioncor2[38] for patched frame motion correction and dose weighting, Gctf[39] was used for estimation of the contrast transfer function parameters, and RELION-2.0[27] for all other image processing steps. Templates for reference-based particle picking were obtained from 2D class averages that were calculated from a manually picked subset of the micrographs. Template-based particle picking was employed to pick particles for both the 100S dimer and 70S monomers. In either case, one round of reference-free 2D classification was used to prune falsely picked particles. For the 100S particle

reconstruction, one round of 3D classification was used where initially a synthetically generated ribosome dimer used as an initial model to identify one class consisting of 12,570 particles that displayed two 70S particles well ordered with respect to each other (Supplementary Table 1).

In the case for the high-resolution reconstruction of the 70S particle, one round of 3D classification was used to remove particles clearly lacking the small subunit and the remaining particles were refined into a single reconstruction initially. As it was apparent that there was further unresolved heterogeneity present in the resulting reconstruction, particularly in the small subunit, fine angular 3D classification was employed to further classify the data. Among classes that clearly lacked the small subunit, we identified one major class that showed good density for both the large and small subunit. This subset of data (224,554 particles) was used for the final reconstruction of the 70S particle.

Reported resolutions are based on the gold-standard Fourier shell correlation (FSC) = 0.143 criterion and FSC curves were corrected for the effects of a soft mask on the FSC curve using high-resolution noise substitution[40]. All 3D refinements were started from a 60 Å low-pass-filtered initial model. Before visualization, all density maps were corrected for the modulation transfer function of the detector and then sharpened by applying a negative B-factor that was estimated using automated procedures[41].

**Model building and refinement**. Model building of 100S of *S. aureus* rRNA and proteins was executed by combining template guided and de novo model building. The previously published structure of *S. aureus* 50S[42] was used as an initial template for the large ribosomal subunit. The structures of *T. thermophilus* 70S[43] were used as an initial template for the small ribosomal subunit. Each model was docked into the proper masked and sharpened maps using UCSF chimera[44], mutated and manually adjusted using COOT[45]. Model refinement was performed using an iterative approach including real space refinement and geometry regularization in COOT, followed by real space refinement using PHENIX[46]. Once the refinement of each ribosomal subunit was completed, two copies of each particle was fitted into the 100S map and further refined.

**Negative stain**. SA70S complex mixture with tRNA and mRNA (3.5 μl of a 0.5 mg ml$^{-1}$ solution) was applied on pre-coated carbon grids for 30 s and then blotted with filter paper. Afterward the grid was wiped on water drop and three times on 2% uranyl acetate drop, blotted with filter paper in between.

**Size exclusion chromatography**. Size exclusion chromatography (SEC) was performed on a Superdex-200 Increase 10/300 GL column (GE Healthcare) equilibrated with SEC buffer (20 mM Tris (pH 7.9), 20 mM MgCl$_2$, 250 mM NH$_4$Cl, 10 mM KCl) at a flow rate of 0.5 ml min$^{-1}$. Fifty micrograms of purified HPF$_{SA}$ proteins were used for analysis. Absorbance at 280 nm was used to detect proteins eluted from the column. Apparent molecular weights were estimated using a standard curve of the molecular masses of protein markers (BioRad): myoglobin (17 kDa), ovalbumin (44 kDa), IgG bovine (158 kDa), and thyroglobulin (670 kDa).

**Acute heat-killing assay**. Stationary phase TSB cultures (supplemented with 5 μg ml$^{-1}$ chloramphenicol) of *S. aureus hpf*-null derivatives were diluted with TSB to an OD$_{600}$ of 0.37–0.38, split into three portions and incubated at 37, 58, and 62 ° C in a thermomixer. After 30 min incubation, cells were subject to 10-fold serial dilutions with the sterile 1 × PBS buffer. Five microliters of each dilution was spotted on TSB agar plates and incubated for 18 h at 37 °C. To enumerate CFU ml$^{-1}$, aliquots (75 μl) of serially diluted cells were spread onto TSB agar plates and values of CFU ml$^{-1}$ were determined after 1 day incubation at 37 °C. Two to three independent experiments were performed on separate day.

**Data availability**. Cryo-EM maps have been deposited in the Electron Microscopy Databank with accession code EMDB-3637 with PDB ID code 5NG8 and EMDB-3640 with PDB ID code 5NGM. The additional data that support the findings of this study are available from the corresponding authors upon request.

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

## Acknowledgements

The data were collected at the Cryo-EM Swedish National Facility. We thank Dr David Wood at the SLU Protein Core for assistance with SEC analysis, M. Carroni and J. Conrad for help with data collection, S. Fleischmann for computing support, Dr H. Rozenberg for fruitful discussions, and Y. Halfon and Z. Eyal S. Tel-Or for experimental support. Funding was provided by the FEBS Long-Term Fellowship (SA), Swedish Research Council (NT_2015-04107), Swedish Foundation for Strategic Research (Future Leaders Grant FFL15-0325), Ragnar Soderbergs foundation (Fellowship in Medicine M44/16), Raymond and Beverly Sackler Churchill college fellowship (A.A.); European Research Council Grant 322581; the Kimmelman Center for Macromolecular Assemblies; A.E.Y. holds the Martin S. and Helen Kimmel Professorial Chair at the Weizmann Institute of Science. M.-N.F.Y. is supported by the PEW Charitable Trusts, the Edward Mallinckrodt Jr. Foundation, and NIH GM121359.

## Author contributions

M.-N.F.Y. and A.B. constructed the *S. aureus* strains and performed SEC. D.M. cultured the cells and purified the 100S ribosomes. S.A. collected and processed the cryo-EM data. D.M. performed the coordinate refinement of the atomic model. D.M. and S.A. interpreted the structure. A.A. supervised sample preparation, the EM analysis and overall project design and execution. A.B., E.Z., and A.E.Y. supervised model building and refinement and overall project design and execution. D.M., S.A., B.A., and A.A. wrote the manuscript and all authors have commented on its final version.

## Additional information

**Competing interests:** The authors declare no competing financial interests.

