## [Peer Review File · Nature Communications]

Reviewers' comments:

Reviewer #1 (Remarks to the Author):

In the study by Donna et al., the authors provide structural and functional characterization of hibernation promoting factor (HPF) in stalling and dimerization of *Staph. aureus* ribosomes. The manuscript presents cryo-EM structures that depict the particle arrangement of HPF binding to the 30S subunit of two 70S ribosomes to form a 100S particle. The authors further characterize individual (N and C) domains of HPF-SA and show that the NTD is responsible for binding to the ribosome, while the CTD homodimerizes. Finally, the authors include an enlightening analysis that compares their new data with what is known with similar factors previously characterized for *E. coli* ribosomes. Together, the data provide additional evidence for understanding a controlled mechanism for stalling translation that is common, yet subtly different, in diverse bacteria. The findings should be of interest to the general audience of *Nature Communications*. That said, clarification of the following points would improve the manuscript further.

In the report the authors use, essentially, two refined reconstructions for interpreting their results: one focuses on the 100S dimer with imposed C2 symmetry that refines to 6.7Å and the other is obtained from the 70S monomer, which refines to ~3Å resolution. It is not clear from the text, whether the 70S refinement is taken from all 70S particles or from only those that are in the context of the 100S particle exclusively. This is an important distinction as interpretation of rRNA helix and/or subunit (30S head) conformational changes are attributed to binding of HPFSA and or dimerization. One of several examples is in the section titled "100S stabilization achieved by HPF induces conformation changes of the 30S subunit". Is this heterogeneity due to differences in occupancy of HPFSA? Along these lines, if there is indeed conformational heterogeneity within a homogenous set of 100S particles, then forcing C2 symmetry would partially dilute the structural differences that exist, thereby limiting interpretation of structural/conformational heterogeneity.

Related to the previous point, it is not clear why the authors chose to interpret the lower resolution 100S, C2-imposed reconstructions at all. The forced symmetry operation could include density that is an artifact of averaging heterogeneity and, therefore, not optimal for modeling. The complete density for HPFSA should be observed in the 70S refined structures, as long as the populations of particles all come from 100S dimers.

Related to the above two points, what fraction of ribosomes are in 70S versus 100S fractions? Does overexpression of HPF in an HPF-null line have any consequences on cell growth?

The authors' inclusion of supporting biochemical data examining HPF-CTD and HPF-NTD was a nice addition to the structural work. For a better understanding of HPF and RMF-like functions, it would be beneficial if the authors could include the cellular fate of those two constructs as well. Is the HPFSA-NTD (like *E. coli* RMF) sufficient to completely or partially stall translation? What about the HPFSA-CTD?

Related to the previous point, the authors only superficially discuss the functional need for dimerization of HPFSA and 100S ribosome formation to stall translation. An expanded analysis of the HPFSA-CTD and HPFSA-NTD data would help in interpreting the need for dimerization and would help in understanding bacterial translational control through ribosome stalling.

Minor comments:

Several 'typos' were noted throughout the manuscript.

Figure 1 is difficult to interpret and should be improved for clarity: It would help if the 30S and 50S models were shown in different colors. What is contributing to the extra density of the 50S subunit in to top left panel. 'protuberance' probably should not be in italics.

Line 87 and Figure 2: "physically blocking the binding pocket of mRNA and tRNA..": This is a potentially misleading figure as it would appear that tRNA molecules are in the HPF-100S structure, which is not the case. This point could be further clarified that the mRNA-tRNA are not in the structure, but only modeled into it. Also, removing the density cloud around the tRNA cartoon/ribbons in the figure would help in understanding that the density is not seen in the structure.

Figure 2 legend: "...R95 forms hydrogen bonds with G891 and Y47 is stacked with the base of G1022..." These are bold statements and even at the resolution of the present cryo-EM reconstruction would be difficult to support. It is recommended that density supporting these claims be included or the wording changed to be more conservative regarding the accuracy and precision of the fitted models.

Reviewer #2 (Remarks to the Author):

In the presented work by Matzov et al., the authors obtained high-resolution Cryo-EM reconstruction of the 100S ribosome dimer from the Gram-positive bacterium *Staphylococcus aureus* that has a long version of the Hibernation Promoting Factor. The process of ribosome inactivation through 100S formation attracts lots of attention recently, primarily because it plays pivotal role in survival of bacteria during harsh conditions including treatment with antibiotics and is believed to contribute to the development of drug resistance. The most interesting finding of this study is that dimerization of ribosomes in *S. aureus* occurs via different mechanism than the one reported for *E. coli*, in which two hibernation factors RMF and HPF cause conformational changes of the small subunit, so that ribosomes become more prone to adhere to each other and form dimers. In *S. aureus* the C-terminal domain of the homologous factor HPF directly mediates dimerization. Also, the structures obtained in this work reveal that the 70S-70S interfaces in the 100S dimers from *E. coli* and *S. aureus* are conceptually different pointing to species-specificity of ribosome dimerization.

In summary, the authors shared an interesting discovery of the alternative mechanism of ribosome dimerization in *S. aureus*. However, after reading this manuscript I have mixed feelings. On one hand, the authors obtained a very high-quality structures with extremely good resolution among cryo-EM structures that allowed them to build accurate structural models. Also, the manuscript is very concise and well-written (with some minor typos and glitches that are pointed below). On the other hand, the biological significance of the obtained structures is poorly discussed, making the manuscript (in its current form) more a description of the structures, rather than a complete story, in which structural features have their assigned biological roles. This work provides important results and if the main critical points denoted below are addressed by the authors, this manuscript could be potentially considered for publication.

Comments, suggestions and questions to the authors:

Major critical points:

1. Perhaps, the most exciting part of this work is the contact between the two C-terminal domains of the two HPF proteins bound to the two 70S ribosomes. And yet, the authors choose not to elaborate much on this important structural feature. In my opinion, this is one of the most important findings of

this work. It would be really curious to know how exactly the two CTDs of HPF interact with each other? What kind of contacts they establish? What makes that interaction strong, so that this contact can keep two 70S "giants" together? Also, this interaction of the CTDs needs to be illustrated, either as additional panels in Figure 3, or as a separate figure.

2. Although authors clearly illustrate in Figure 2C that NTD of the long HPF clashes with all three tRNAs and mRNA on the 30S subunit, they have only couple short sentences about this in the main text, which is very easy to miss. I would like to suggest to discuss in more detail this structural aspect of the HPF binding site in relation to its biological function – inactivation of the ribosomes. In the current form, the manuscript implies that the reader is very well-versed in this topic. For the broad readership of NCOMM this material needs to be better digested and explained with more details. Also, figure 2 could have been done in a more thoughtful way – currently it looks like a set of rather disconnected panels (each of which makes sense only by itself).

3. This is entirely up to the authors, but also I would like to suggest to merge the "Results" and "Discussion" sections together, so that results are being discussed immediately. In my opinion, this particular work could benefit from such a conversion.

Minor comments and critical points:

1. References currently are not in Nature Communications format.
2. Page 1, Line 27: Authors might wish to consider changing "...anticodon region of tRNA..." to "...anticodon regions of A- and P-site tRNAs..." to emphasize that presence of the HPF is incompatible with any of the tRNAs.
3. Page 1, Line 29: Correct "...HPFec..." to "...RMFec..."
4. Page 2, Line 33: Remove dash from latin "in vitro" and change "...electron cryo-microscopy..." to "...cryo-electron microscopy...".
5. Page 2, Line 42: Correct the reference.
6. Page 2, Line 53: Change "...was found block..." to "...was found to block...".
7. Page 2, Line 56: Correct the word "species"
8. Page 2, Last sentence of the Intro: The statement about therapeutic target made in this sentence is questionable and is largely a speculation, because currently it is very hard to imagine. I would like to suggest to keep the first part of the sentence and remove the second part after the comma.
9. Page 3, Line 84: Change "The 100 residues..." to "The first 100 residues..."
10. Page 3, The last sentence in the paragraph on lines 89-91: It is unclear, why authors were looking for the electron density in their structure around the E. coli RMF binding pocket, since S.aureus does not even have RMF? This needs to be explained in the text.
11. Page 4, First paragraph: The authors are describing their structure and stating that "H26 of one ribosome interacts with uS2 of the other and vice versa". Obviously, this interaction can happen even without HPF, so why dimers are not formed in the absence of HPF? Although the answer to this question might be obvious to some readers, it needs to be explained for the broad readership.
12. Page 4, Paragraph on lines 105-111: The authors describe a very interesting experiment in which they deleted the entire N-terminal domain of the HPF and still see the dimerization of the 70S ribosomes. Then why do we need the entire HPF protein? Again, the answer might be obvious to some readers, but would be better if explained and discussed.
13. Page 5, Line 137: The authors pointed to the observation that in EC100S there are two main contacts between the two 30S particles in the 100S dimer: between the heads and between the bodies. The first contact involves H39 and two proteins uS9 and uS10, while the second contact involves only ribosomal proteins – uS2 of one particle inserts into the cavity formed by uS3, uS4, and uS5 of the second particle. Currently in the text these parts are mixed up in couple of places.
14. Page 5, Line 146: See previous comment and insert uS4 after uS3.

We are most grateful to the positive constructive comments of both reviewers that improved our manuscript significantly, making it more accessible to the audience of Nature Communications. We believe we addressed all of the reviewers concerns in the revised manuscript (see below):

Reviewer #1:

1. In the report the authors use, essentially, two refined reconstructions for interpreting their results: one focuses on the 100S dimer with imposed C2 symmetry that refines to 6.7Å and the other is obtained from the 70S monomer, which refines to ~3Å resolution. It is not clear from the text, whether the 70S refinement is taken from all 70S particles or from only those that are in the context of the 100S particle exclusively. This is an important distinction as interpretation of rRNA helix and/or subunit (30S head) conformational changes are attributed to binding of HPFSA and or dimerization. One of several examples is in the section titled “100S stabilization achieved by HPF induces conformation changes of the 30S subunit”. Is this heterogeneity due to differences in occupancy of HPFSA? Along these lines, if there is indeed conformational heterogeneity within a homogenous set of 100S particles, then forcing C2 symmetry would partially dilute the structural differences that exist, thereby limiting interpretation of structural/conformational heterogeneity.

Related to the previous point, it is not clear why the authors chose to interpret the lower resolution 100S, C2-imposed reconstructions at all. The forced symmetry operation could include density that is an artifact of averaging heterogeneity and, therefore, not optimal for modeling. The complete density for HPFSA should be observed in the 70S refined structures, as long as the populations of particles all come from 100S dimers.

Authors Response: Although the sample used to acquire the dataset for the 3Å reconstruction of the 70S was the same as the sample for the 100S, and that the particles that were used for the refinement all contain HPF and therefore part of 100S dimer, not all 100S dimers are equal. In particular, it was noticeable that for the 100S maps, large numbers of particles displayed high degree of flexibility between the two 70S particles, and only one class had a stable arrangement of the two 70S ribosomes. Therefore the dimerization interface is blurred in our 70S reconstruction because the refinement is driven largely by the signal of the 70S particle and that the 70S reconstruction contains particles that would have been classified separately in the context of 100S. We therefore think it is important to present both reconstructions where we used the 70S reconstruction to build the best model of the ribosome involved in dimerization, whereas the 100S reconstruction is necessary to observe the interactions involved between the two units. To emphasize that the C2 applied reconstruction is representative of the asymmetric reconstruction we had added another supplementary figure (figure S4) with a side-by-side view of the two maps.

2. Related to the above two points, what fraction of ribosomes are in 70S versus 100S fractions? Does overexpression of HPF in an HPF-null line have any consequences on cell growth?

Authors Response: As presented in Basu & Yap, NAR 2016 (cited throughout the manuscript) the ratio between 70S and 100S is >40%. Also, under the growth conditions that were used for this paper overexpression of HPF does not cause growth arrest. We believe that adding more information about cells fate under different conditions that are not related to this manuscript might confuse the reader.

3. The authors' inclusion of supporting biochemical data examining HPF-CTD and HPF-NTD was a nice addition to the structural work. For a better understanding of HPF and RMF-like functions, it would be beneficial if the authors could include the cellular fate of those two constructs as well. Is the HPFSA-NTD (like E. coli RMF) sufficient to completely or partially stall translation? What about the HPFSA-CTD?

Authors Response: HPF does not act to stall ribosome during translation. HPF-NTD binding prevents the entry of tRNA and mRNA and thereby inhibiting translation initiation. Since that point was not clear we added lines 90-98:

“The first step of the prokaryotic protein biosynthesis initiation stage involves the binding of a specific initiator methionyl tRNA and the mRNA to the small ribosomal subunit. The large ribosomal subunit then joins the complex, forming a functional ribosome on which elongation of the polypeptide chain proceeds. The adaptor function of the tRNAs involves two separated regions of the molecule. All tRNAs possess CCA at their 3' terminus, and amino acids are covalently attached to the ribose of the terminal adenosine. The mRNA template is recognized by the anticodon loop, located at the other end of the tRNA, which binds to the appropriate codon by complementary base pairing. Hence, blocking the binding pockets of the mRNA and tRNA molecules inhibits the translation process.”

Like in the WT strain, both truncated mutants do not cause growth arrest under the growth conditions that were used for this paper.

4. “Related to the previous point, the authors only superficially discuss the functional need for dimerization of HPFSA and 100S ribosome formation to stall translation. An expanded analysis of the HPFSA-CTD and HPFSA-NTD data would help in interpreting the need for dimerization and would help in understanding bacterial translational control through ribosome stalling.”

Authors Response: Basu & Yap, (NAR 2016) have showed that in *in vitro* dimerization assays, purified HPF-NTD by itself does not form 100S ribosome and is severely impaired in ribosome binding, whereas in the current manuscript we showed that HPF-CTD is a dimer in solution and on the ribosome. Since HPF_{SA} has dual functionality and both of its domains can function independently, the “need for dimerization” might be difficult to determine. We did add to the discussion our speculation (lines 205-209): “Bacterial translation and transcription are coupled and the binding sites (e.g. uS2) of RNA polymerase (RNAP) on the 70S ribosome [31] coincidentally overlap with the 70S dimerizing interface. Therefore one might speculate that homo-dimerizing 70S prevent spurious interactions between the RNAP and ribosome in the crowded cytoplasm when translation should not be occurring.”

5. “Figure 1 is difficult to interpret and should be improved for clarity: It would help if the 30S and 50S models were shown in different colors. What is contributing to the extra density of the 50S subunit in to top left panel. ‘protuberance’ probably should not be in italics.”

Authors Response: Figure 1 and its legend were changed. The extra density belonged to a part of an rRNA helix that at first we decided to omit due to the limited resolution at this area. Now seeing that omitting this density may raise questions, we used the original model in the new figure.

6. “Line 87 and Figure 2: “physically blocking the binding pocket of mRNA and tRNA.”: This is a potentially misleading figure as it would appear that tRNA molecules are in the HPF-100S structure, which is not the case. This point could be further clarified that the mRNA-tRNA are not in the structure, but only modeled into it. Also, removing the density cloud around the tRNA cartoon/ribbons in the figure would help in understanding that the density is not seen in the structure.”

Authors Response: Figure 2.B and its legend were altered and the following sentences were added to the text for further clarification in line 87-89:

“Superposition of Tth70S complex with mRNA and three tRNAs at their binding pockets (PDB code 4W2F) on one of the 70S ribosome composing the SA100S dimer presented here, demonstrates that N-HPF_{SA}”.

7. “Figure 2 legend: “..R95 forms hydrogen bonds with G891 and Y47 is stacked with the base of G1022...” These are bold statements and even at the resolution of the present cryo-EM reconstruction would be difficult to support. It is recommended that density supporting these claims be included or the wording changed to be more conservative regarding the accuracy and precision of the fitted models.”

Authors Response: Figure 2.A was altered and a density map was added.

Reviewer #2:

1. “Perhaps, the most exciting part of this work is the contact between the two C-terminal domains of the two HPF proteins bound to the two 70S ribosomes. And yet, the authors choose not to elaborate much on this important structural feature. In my opinion, this is one of the most important findings of this work. It would be really curious to know how exactly the two CTDs of HPF interact with each other? What kind of contacts they establish? What makes that interaction strong, so that this contact can keep two 70S “giants” together? Also, this interaction of the CTDs needs to be illustrated, either as additional panels in Figure 3, or as a separate figure.”

Authors Response: We addressed this comment both in the Results and in the Discussion sections: lines 117-122: “Due to the limited resolution of the cryo-EM maps in this region the specific interactions between the two C-HPFSA domains that allows self-dimerization could not be characterized. However the homology model suggests a three-dimensional domain-swapping mechanism in which the beta strand of one C-HPFSA is packed in parallel against the antiparallel β -sheet of its dimer counterpart (Fig. 3C). The fact that the intra beta strand interaction is parallel to the beta sheet makes the dimer less stable.”

Lines 185-194: “The 30S-30S interactions are mediated by a homo-dimer of C-terminal domains from two HPF_{SA} which allows for rRNA and ribosomal proteins to interlock and further secure the connection between the two 70S ribosomes. Such dimerization presents two recognition sites for rRNA binding, in this case h26 form both 30S subunits and can therefore provide a cooperative interaction that strengthens the affinity of the C-terminal domain for the rRNA [31]. . Notably, although the interactions between the two 70S should be strong, they certainly are not rigid, since the 3D classification shows that the large majority of the dimers exhibit flexibility between to two ribosomes. Indeed the dimer stabilizing parallel beta sheet interactions facilitate weaker dimer interaction (compared to an antiparallel arrangement).”

Also, an additional panel was added to Figure 3 describing the CTD of HPF_{SA}.

2. Although authors clearly illustrate in Figure 2C that NTD of the long HPF clashes with all three tRNAs and mRNA on the 30S subunit, they have only couple short sentences about this in the main text, which is very easy to miss. I would like to suggest to discuss in more detail this structural aspect of the HPF binding site in relation to its biological function – inactivation of the ribosomes. In the current form, the manuscript implies that the reader is very well-versed in this topic. For the broad readership of NCOMM this material needs to be better digested and explained with more details. Also, figure 2 could have been done in a more thoughtful way – currently it looks like a set of rather disconnected panels (each of which makes sense only by itself).

Authors Response: Figure 2 and its legend were altered and the following sentences were added to the text for further clarification in line 87-89:

“Superposition of Tth70S complex with mRNA and three tRNA at their binding sites (PDB code 4W2F) on one of the 70S ribosome composing the SA100S dimer presented here demonstrates that

N-HPF_{SA}”.

The addition of lines 90-98 would also clarify this point (See reply the reviewer #1 3rd comment).

3. This is entirely up to the authors, but also I would like to suggest to merge the “Results” and “Discussion” sections together, so that results are being discussed immediately. In my opinion, this particular work could benefit from such a conversion.

Authors Response: We respectfully decline this suggestion and would like to keep the manuscript at its current format.

4. References currently are not in Nature Communications format.

Authors Response: If we understood correctly, the proper formatting will be imposed upon publication. “There is no need to spend time visually formatting the manuscript: *Nature Communications* style will be imposed when the paper is prepared for publication.”

5. Page 1, Line 27: Authors might wish to consider changing “...anticodon region of tRNA...” to “...anticodon regions of A- and P-site tRNAs...” to emphasize that presence of the HPF is incompatible with any of the tRNAs.

6. Page 1, Line 29: Correct “...HPFec...” to “...RMFec...”

7. Page 2, Line 33: Remove dash from latin “in vitro” and change “...electron cryo-microscopy...” to “...cryo-electron microscopy...”.

8. Page 2, Line 42: Correct the reference.

9. Page 2, Line 53: Change “...was found block...” to “...was found to block...”.

10. Page 2, Line 56: Correct the word “species”

11. Page 2, Last sentence of the Intro: The statement about therapeutic target made in this sentence is questionable and is largely a speculation, because currently it is very hard to imagine. I would like to suggest to keep the first part of the sentence and remove the second part after the comma.

12. Page 3, Line 84: Change “The 100 residues...” to “The first 100 residues...”

Authors Response: Comments 5-12 were corrected as suggested by reviewer #2

13. Page 3, The last sentence in the paragraph on lines 89-91: It is unclear, why authors were looking for the electron density in their structure around the E. coli RMF binding pocket, since S.aureus does not even have RMF? This needs to be explained in the text.

Authors Response: lines 99-101 were added to clarify this issue: “RMF_{EC} also shares a similar degree of identity with the middle part of HPF_{SA} therefore it was presumed that S. aureus long HPF might be a hybrid of the short HPF and RMF.”

14. Page 4, First paragraph: The authors are describing their structure and stating that “H26 of one ribosome interacts with uS2 of the other and vice versa”. Obviously, this interaction can happen even without HPF, so why dimers are not formed in the absence of HPF? Although the answer to this question might be obvious to some readers, it needs to be explained for the broad readership.

Authors Response: See reply to 1st comment of reviewer #2

15. Page 4, Paragraph on lines 105-111: The authors describe a very interesting experiment in which they deleted the entire N-terminal domain of the HPF and still see the dimerization of the 70S ribosomes. Then why do we need the entire HPF protein? Again, the answer might be obvious to some readers, but would be better if explained and discussed.

Authors Response: The NTD of HPF_{SA} is responsible for the inhibition of the translation process. The CTD of HPF_{SA} is directly involved in dimer formation. Both domains can function independently from the other domain. Other than describing the dual functionality of this protein in the results section (lines 82-132), it was also summarized both in the introduction (lines 50-53) and discussion sections (lines 201-204).

16. Page 5, Line 137: The authors pointed to the observation that in EC100S there are two main contacts between the two 30S particles in the 100S dimer: between the heads and between the bodies. The first contact involves H39 and two proteins uS9 and uS10, while the second contact involves only ribosomal proteins – uS2 of one particle inserts into the cavity formed by uS3, uS4, and uS5 of the second particle. Currently in the text these parts are mixed up in couple of places.

17. Page 5, Line 146: See previous comment and insert uS4 after uS3.

Authors Response: There are currently two independent cryo-EM maps of *E.coli* 100S dimers. In both studies two different interpretations in regards of the participating moieties were proposed. For the sake of completeness we chose to refer to both studies. We realize that this might be confusing so lines 154-155 were altered so that this point might become clearer. Protein uS4 is not mentioned in the text.

Reviewers' comments:

Reviewer #1 (Remarks to the Author):

Despite the reasonable explanation in the critique, the revised manuscript continues to present confusion with regards to the processing of the data. In the first section of the results, the authors state that the "2D class averages suggested that the connection between the two 70S ribosomes is generally flexible", which was confirmed by subsequent 3D sorting (the flexibility is further discussed in the 'Discussion'). However, later in the same section of results, the authors state there was "a clear axis of two-fold symmetry within the 100S particle", so they applied C2 symmetry to improve the resolution. These two statements contradict each other. I suspect that there is a pseudo-two-fold axis of symmetry and treating it as a true two-fold axis is not appropriate in this circumstance. Therefore the second half (imposed symmetry refinement) of this paragraph should be omitted from the study.

The questionable application of C2 symmetry occurs again on lines 104-106 to interpret the interface of the 30S ribosome with HPFSA dimer. If, as the authors state, the connection is flexible, then C2 would improperly smear the density of the refined structure.

It is unclear why the authors chose to entirely remove the description of the complementary functional studies from the revised draft. In fact, the biochemical data (Figs S7D-H) are no longer cited or even mentioned in the revised manuscript.

Related to the point above, and contrary to the authors response to the previous critique, this reviewer feels that a discussion of the fates of cells lacking the N- or C-terminal domain of HPFSA would only strengthen the study instead of, as the authors claim in the rebuttal, "confuse the reader." The entire article focuses on the mechanism of HPF-SA and so the "information about cell fate under different conditions" (where different conditions address the role of the HPFSA N- and C-termini) are absolutely related to the manuscript and should therefore be included/discussed.

Reviewer #2 (Remarks to the Author):

After carefully reading the revised version of the manuscript, I need to say that the authors done an excellent job revising the manuscript! In my opinion, most of the critical points raised by this reviewer were satisfactorily addressed. The manuscript definitely became clearer after the revisions and most of the confusing points were removed. Especially, I liked the inclusion of the sentence in the "discussion" section speculating that the dimerization itself might be required to prevent RNA polymerase from docking to the ribosome during stationary phase. In fact, this idea is so great that, in my opinion, the authors might wish to consider including a sentence on that in the abstract. Also, the clarity of the figures 1,2 and 3 has been substantially increased after modifications asked by the other reviewer.

In summary, manuscript in its current form could be accepted for publication.

We thank again the reviewers for careful and thorough reading of this manuscript and for the thoughtful comments and constructive suggestions, which helped to improve the quality of this manuscript. The corresponding changes and refinements made in the revised paper are summarized below:

Reviewer #1 (Remarks to the Author):

Despite the reasonable explanation in the critique, the revised manuscript continues to present confusion with regards to the processing of the data. In the first section of the results, the authors state that the “2D class averages suggested that the connection between the two 70S ribosomes is generally flexible”, which was confirmed by subsequent 3D sorting (the flexibility is further discussed in the ‘Discussion’). However, later in the same section of results, the authors state there was “a clear axis of two-fold symmetry within the 100S particle”, so they applied C2 symmetry to improve the resolution. These two statements contradict each other. I suspect that there is a pseudo-two-fold axis of symmetry and treating it as a true two-fold axis is not appropriate in this circumstance. Therefore the second half (imposed symmetry refinement) of this paragraph should be omitted from the study.

The questionable application of C2 symmetry occurs again on lines 104-106 to interpret the interface of the 30S ribosome with HPFSA dimer. If, as the authors state, the connection is flexible, then C2 would improperly smear the density of the refined structure.

Authors Response: We admit that the two mentioned statements in the first section of Results can be interpreted as contradicting each other. We therefore clarified this section (lines 64-70), which now states that a single 3D class showed a well defined orientational state for both 70S ribosomes. This validates applying the two-fold symmetry and should eliminate possible misinterpretations.

It is unclear why the authors chose to entirely remove the description of the complementary functional studies from the revised draft. In fact, the biochemical data (Figs S7D-H) are no longer cited or even mentioned in the revised manuscript.

Authors Response: The observations of the reviewer are exact. Figs S7D-E was mislabeled as Figs S5D-E in the previous revision. This was corrected and the functional analysis is now cited. The citation has been corrected and a description of HPF_{SA} self-dimerization was added in lines 128-129.

Related to the point above, and contrary to the authors response to the previous critique, this reviewer feels that a discussion of the fates of cells lacking the N- or C-terminal domain of HPFSA would only strengthen the study instead of, as the authors claim in the rebuttal, “confuse the reader.” The entire article focuses on the mechanism of HPF-SA and so the “information about cell fate under different conditions” (where different conditions address the role of the HPFSA N- and C-termini) are absolutely related to the manuscript and should therefore be included/discussed.

Authors Response: In response to the reviewer’s comments, we have now included additional heat stress experiments (Fig. S8), which show that the dimerization function of HPF_{SA} is required for cell

viability upon acute heat killing. NTD-HPF without dimerizing activity is subject to cell death at the same rate as the vector control strain. We also included additional narratives on the potential role of dimerization in ribosome integrity (see lines 129-132 and 213-217).

REVIEWERS' COMMENTS:

Reviewer #1 (Remarks to the Author):

The authors have sufficiently addressed all concerns that arose from the prior submissions. In this reviewer's opinion, the revised article is better communicated and interpreted and the results would be of general interest to the readers of Nature Communications.